# A Dengue Virus Type 2 (DENV-2) NS4B-Interacting Host Factor, SERP1, Reduces DENV-2 Production by Suppressing Viral RNA Replication

**DOI:** 10.3390/v11090787

**Published:** 2019-08-27

**Authors:** Jia-Ni Tian, Chi-Chen Yang, Chiu-Kai Chuang, Ming-Han Tsai, Ren-Huang Wu, Chiung-Tong Chen, Andrew Yueh

**Affiliations:** 1Institute of Biotechnology and Pharmaceutical Research, National Health Research Institutes, Miaoli 35053, Taiwan; 2Department of Life Sciences, National Central University, Jhongli 32001, Taiwan; 3Division of Animal Technology, Agricultural Technology Research Institute, Miaoli 35053, Taiwan; 4Institute of Microbiology and Immunology, National Yang-Ming University, Taipei 11221, Taiwan

**Keywords:** dengue virus, nonstructural protein 4B, SERP1, endoplasmic reticulum stress, viral life-cycle

## Abstract

Host cells infected with dengue virus (DENV) often trigger endoplasmic reticulum (ER) stress, a key process that allows viral reproduction, without killing the host cells until the late stage of the virus life-cycle. However, little is known regarding which DENV viral proteins interact with the ER machinery to support viral replication. In this study, we identified and characterized a novel host factor, stress-associated ER protein 1 (SERP1), which interacts with the DENV type 2 (DENV-2) NS4B protein by several assays, for example, yeast two-hybrid, subcellular localization, NanoBiT complementation, and co-immunoprecipitation. A drastic increase (34.5-fold) in the SERP1 gene expression was observed in the DENV-2-infected or replicon-transfected Huh7.5 cells. The SERP1 overexpression inhibited viral yields (37-fold) in the DENV-2-infected Huh7.5 cells. In contrast, shRNAi-knockdown and the knockout of SERP1 increased the viral yields (3.4- and 16-fold, respectively) in DENV-2-infected HEK-293 and Huh7.5 cells, respectively. DENV-2 viral RNA replication was severely reduced in stable SERP1-expressing Huh7.5 cells transfected with DENV-2 replicon plasmids. The overexpression of DENV-2 NS4B alleviated the inhibitory effect of SERP1 on DENV-2 RNA replication. Taking these results together, we hypothesized that SERP1 may serve as an antiviral player during ER stress to restrict DENV-2 infection. Our studies revealed novel anti-DENV drug targets that may facilitate anti-DENV drug discovery.

## 1. Introduction

DENV infection is a common arbovirus illness in humans [1]. DENV has four serotypes (one to four) and belongs to the genus *Flavivirus* within the family Flaviviridae. In addition to DENV, other viruses in the genus *Flavivirus*, such as the West Nile virus (WNV), Japanese encephalitis virus (JEV), yellow fever virus, and tick-borne encephalitis virus, are pathogens of humans and animals. After acute dengue viral infection, patients often develop fever, rash, headache, muscle and joint pain, and nausea [2]. The most severe form of the disease is dengue hemorrhagic fever/dengue shock syndrome, which is associated with vascular leakage, hemorrhage, and shock [3]. More than 3.5 billion people are at risk for DENV infection. It has been estimated that each year, up to 96 million of the 390 million DENV infections in the world are symptomatic [1]. 

The *Flavivirus* genome is a single-stranded, positive-sense RNA of approximately 11 kb in length. Its genomic RNA consists of a 5′ untranslated region (UTR), an open reading frame (ORF), and a 3′ UTR [4]. The ORF encodes a polyprotein that is processed into three structural proteins (capsid (C), premembrane (prM), and envelope (E)) and seven nonstructural proteins (NS1, NS2A, NS2B, NS3, NS4A, NS4B, and NS5) [5,6,7]. The structural proteins are the main components of the virion. The nonstructural proteins are important for viral RNA replication [8].

Flavivirus NS4B is essential for viral RNA replication [9,10,11] and the evasion of the host immune response [12,13]. NS4B, 27 kDa, is an integral membrane protein that has five predicted transmembrane domains (TMD) [14]. The pTMD4 of the NS4A region (2K) serves as a signal sequence for the translocation of the adjacent NS4B into the endoplasmic reticulum (ER) lumen [15]. DENV NS4B is one of the membrane-bound viral replication complexes in the ER [14]. The DENV NS4B expression was shown to modulate innate immunity signaling by blocking the α/β interferon pathway through the inhibition of STAT1 phosphorylation, dimerization, and translocation to the nucleus [12,13]. Furthermore, NS4B is able to induce mitochondrial elongation and inhibit the activation of the immune response elicited by the mitochondrial antiviral-signaling protein-dependent interaction of the ER with mitochondria to promote infection [16]. The DENV-2 NS4B protein expression also modulates the unfolded protein response (UPR) transcriptional activation. The expression of the DENV NS4B protein suppressed the induction of heavy-chain binding protein (BiP/GRP78) and EDEM in the unfolded protein response (UPR) [17]. Little is known regarding the interaction between NS4B and the host factors to support the virus life-cycle. In this study, we identified the interaction of stress-associated endoplasmic reticulum protein 1 (SERP1) with dengue virus type 2 (DENV-2) NS4B by a membrane-based split-ubiquitin yeast two-hybrid system.

The detailed function of SERP1 in the cells is not clear. SERP1, also known as ribosome-associated membrane protein 4 (RAMP4) [18], is a tail-anchored protein with an N-terminus exposed on the cytoplasmic (residues 1–115), and a C-terminus inserted on the luminal side of the ER membrane (residues 116–198) [19]. SERP1 was reported to interact with subunits (Sec 61α and Sec 61β) of the translocon [18], which acts as a channel and mediates the translocation of polypeptides across membranes to aid protein synthesis [20]. Thus, SERP1 is implicated in the regulation of membrane protein biogenesis. SERP1 overexpression caused the newly synthesized integral membrane proteins to degrade, and then facilitated protein glycosylation in order to protect cells from ER stress [21]. The genetic ablation of SERP1 showed that the SERP1^−/−^ mice had growth retardation, increased mortality, impaired glucose tolerance, and ER stress. In the pituitary, the increased activation of molecules associated with ER stress (P-eIF2α) and apoptosis (C/EBP homologous protein and caspase 3) resulted in a higher mortality in the SERP1^−/−^ mice than in wild-type (WT) mice [22].

An increased production of DENV proteins during dengue infection leads to the accumulation of misfolded and unfold proteins in the ER [23]. This accumulation results in ER stress and activation of the UPR as a host response to alleviate ER stress. Tunicamycin, thapsigargin, and dithiothreitol induce ER stress and trigger three initiation branches of the UPR, inositol-requiring protein 1 (IRE1), protein kinase RNA-like ER kinase (PERK), and activating transcription factor 6 (ATF6) [24,25]. Similarly, DENV infection induces these three branches of the UPR [26,27] at different infectious stages during ER stress [28]. The PERK arm is activated early in DENV infection, followed by IRE1-XBP1 mid-infection, and ATF6 later during the infection. DENV has developed a strategy to manipulate the host UPR pathways in order to enhance its survival and viral replication for the successful completion of the virus life-cycle.

In this study, we identified a cellular protein, SERP1, that interacted with DENV-2 NS4B and was involved in the DENV life-cycle. SERP1 is induced in Huh7.5 cells by either DENV-2 infection or the transfection of DENV replicon. Cells stably overexpressing SERP1 showed a significant decrease in DENV-2 yields and RNA replication capacity. In contrast, the overexpression of NS4B alleviated the inhibitory effect of SERP1 overexpression on virus replication. The results of the present study suggest that SERP1 is an antiviral protein, and that the interaction of DENV-2 NS4B with SERP1 may counteract the antagonistic effect of SERP1 on the DENV-2 life-cycle.

## 2. Materials and Methods

### 2.1. Cell Lines and Dengue Virus Strain

The HEK-293 cells were maintained in Dulbecco’s Modified Eagle Medium (DMEM) supplemented with 10% fetal bovine serum (FBS) at 37 °C in a 5% CO_2_ incubator. Baby hamster kidney (BHK21) clone 15 cells were kindly provided by Dr. Robert Beatty (Department of Molecular and Cell Biology, University of California at Berkeley, Berkeley, CA, USA), and were cultured in alpha-MEM supplemented with 5% FBS at 37 °C in a 5% CO_2_ incubator. The human hepatocyte cell line Huh7.5 cells were maintained in DMEM supplemented with 10% FBS at 37 °C in a 5% CO_2_ incubator. The dengue virus type 2 (DENV-2; strain 16681) was used for the infection of the Huh7.5 and HEK-293 cells.

### 2.2. Identification of Host Factors Interacting with DENV-2 Proteins by a Membrane-Based Split-Ubiquitin Yeast Two-Hybrid System

The membrane-based split-ubiquitin yeast two-hybrid system (MYTH), DUAL membrane technology, Dualsystems Biotech, Schlieren, Switzerland, was applied to screen NS2B, NS4A, and NS4B (DENV-2, strain 16681) against a human liver cDNA library. MYTH was performed as previously described [29]. The cDNAs encoding the NS2B, NS4A, and NS4B proteins were inserted into a yeast expression vector, p-BT3, fused with C-terminal ubiquitin and artificial transcription factor (TF) Lex-VP16 as baits. In addition, the pPR3N prey vector, harboring a human liver cDNA library, was fused with N-terminal ubiquitin as an interactor protein. The *S. cerevisiae* strain NMY32 was co-transformed with baits and prey plasmids so as to detect the protein–protein interactions (PPIs) of integral membrane proteins. On the interaction of the bait and prey, the N-terminal ubiquitin and C-terminal ubiquitin reconstituted a form of ubiquitin, which caused the ubiquitin-specific-proteases to be released downstream of the TF. The TF translocated into the nucleus and activated the auxotrophic reporter genes, allowing yeast growth on a selective medium. Ten-fold serial yeast dilutions were spotted onto nonselective plates (lacking tryptophan and leucine (-Trp–Leu)) and selective plates (lacking tryptophan, leucine, histidine, and adenine (-Trp–Leu–His–Ade)) for the detection of protein–protein interactions.

### 2.3. Measurement of SERP1 Protein Levels in Flag-SERP1-Overexpressing Huh7.5 Cells by Immunoprecipitation Analysis

The lentiviral vector pLKO-AS2 was purchased from the National RNAi Core Facility, Academia Sinica, Taipei, Taiwan. The full-length SERP1 protein with an N-terminal Flag tag was cloned into the lentiviral vector (pLKO-AS2-Flag-SERP1) in order to overexpress the protein. The HEK 293T cells were co-transfected with the lentiviral plasmid (pLKO-AS2-Flag-SERP1) and two helper plasmids, pCMV-ΔR 8.91 and pMD.G, by Lipofectamine 2000 (Invitrogen, California, USA). The cell supernatants containing lentiviruses were harvested 24 and 48 h after transfection. The viral titers were assayed with a Lenti-X p24 Rapid Titer Kit (Clontech, Mountain View, CA, USA). The Huh7.5 cells were infected with lentiviruses (multiplicity of infection (MOI) = 10)) in a medium containing polybrene (8 µg/mL). Three days after infection, the cells were treated with puromycin (Invitrogen) so as to derive a pool of resistant clones.

The stable Huh7.5 cells overexpressing Flag-tagged SERP1 and the cells receiving an empty vector were lysed in a radioimmunoprecipitation assay (RIPA) buffer, and the protein concentration was determined using a Pierce BCA Protein Assay Kit (Thermo Scientific, Rockford, IL, USA). Equal amounts of cell lysates were analyzed by immunoprecipitation using anti-Flag M2 affinity gel (A2220, Sigma, Missouri, USA) for 2 h at 4 °C. The beads were washed with a lysis buffer and were eluted. The proteins were fractionated on 10% sodium dodecyl sulfate-polyacrylamide gel electrophoresis (SDS-PAGE) and were transferred to the nitrocellulose membranes. An immunoblot analysis was performed using anti-Flag antibody (GTX115043, Genetex, CA, USA) and anti-SERP1 antibody (ab130974, Abcam, Cambridge, MA, USA).

### 2.4. Measurement of Viral Protein Levels by Immunoprecipitation Analysis

To express the NS2B and NS4B proteins, the full-length NS2B and NS4B proteins with or without an N-terminal 2K signal peptide were cloned into the lentiviral vector pLKO-AS2. All of the constructs (pLKO-AS2-NS2B-HA, pLKO-AS2-△2K-NS4B-HA, and pLKO-AS2-NS4B-HA) were expressed with the HA tag epitope fused at the carboxyl terminals. The pLKO-AS2-△2K-NS4B-HA plasmid contained the full-length NS4B lacking an N-terminal 2K signal peptide of NS4A. The parental Huh7.5 cells and Flag-SERP1-overexpressing Huh7.5 cells were transfected with pLKO-AS2-NS2B-HA, pLKO-AS2-△2K-NS4B-HA, or pLKO-AS2-NS4B-HA plasmids. Equal amounts of cell lysates were analyzed by immunoprecipitation using anti-HA magnetic beads (88836, Thermo Fisher Scientific, Waltham, MA, USA), and the HA-tagged proteins were detected by Western blot (WB) using the rabbit polyclonal anti-HA antibody (SG77, Thermo Fisher Scientific, Invitrogen).

### 2.5. The Localization of NS4B and SERP1 in the Endoplasmic Reticulum by Confocal Microscopy

The Flag-SERP1-overexpressing Huh7.5 cells were infected with DENV-2 (multiplicity of infection (MOI) of 5). At 3 d.p.i, the Huh7.5 cells were washed once with phosphate-buffered saline (PBS), fixed with 1% paraformaldehyde for 1 h at room temperature, permeabilized with methanol for 15 min at 4 °C, and blocked with 2% horse serum in PBS. Rabbit polyclonal anti-NS4B (GTX103349, GeneTex), Flag-tag mouse antibody (F1804, Sigma), and mouse monoclonal anti-calnexin (M178-3, MBL Life Science, Nagoya, Japan) were used as the primary antibodies. Alexa Fluor 488 anti-mouse IgG antibody and Alexa Fluor^®^ 568 anti-rabbit IgG H&L antibody (Molecular Probes, Invitrogen) were used as the secondary antibodies. Finally, the coverslips were washed extensively and fixed onto slides. The images were captured by a Leica TCS SP5II confocal microscope (CLSM, Heidelberg, Germany). A colocalization analysis was performed using LAS AF software (Leica Microsystems, Germany) in order to analyze the colocalization of the fluorescent signals.

### 2.6. NanoBiT Complementation Assay of SERP1 and NS4B for Protein–Protein Interactions

NanoLuc^®^ Luciferase is divided into two subunits, large BiT (LgBiT; 18 kDa) and small BiT (SmBiT; 1 kDa peptide), which are expressed as fusions in order to target proteins of interest (Promega). The entire regions of NS2B, NS4B, or SERP1 were coupled to Nano-Luc Binary Technology (NanoBiT) vectors (pBiT1.1-C (TK/LgBiT) vector, pBiT2.1-C (TK/SmBiT) vector, pBiT1.1-N (TK/LgBiT) vector, and pBiT2.1-N (TK/SmBiT) vector; Promega, Madison, WI, USA), according to the manufacturer’s instructions. The HEK-293 cells were cultured in DMEM (Sigma-Aldrich) containing 10% FBS at a density of 1 × 10^4^ cells/well in 96-well plates for 24 h. The HEK-293 cells were transfected with a 40 ng LgBiT fusion construct and a 40 ng SmBiT fusion construct using Lipofectamine 2000 (Invitrogen). Twenty-four hours after transfection, the cells were serum-starved for 30 min in 0% FBS in an Opti-MEM I medium, and the relative luminescence unit (RLU) was measured using a Nano-Glo Live Cell Assay System (Promega) and GloMax^®^ Discover System (Promega).

### 2.7. Interaction of SERP1 with NS4B by Co-Immunoprecipitation (Co-IP)

The Huh7.5 cells were transfected with expression constructs for pLKO-AS2-Flag-SERP1, pLKO-AS2-NS2B-HA, or pLKO-AS2-NS4B-HA. At three days post-transfection, the membrane proteins in the Huh7.5 cells were isolated with Mem-PER™ Plus Membrane Protein Extraction Kit, according to the manufacturer’s protocol (89842, Thermo Fisher Scientific, USA). The protein concentration was determined using a Pierce BCA Protein Assay Kit (23225, Thermo Fisher Scientific, USA). Equal amounts of protein complexes were analyzed by immunoprecipitation using anti-Flag M2 affinity gel (A2220, Sigma) or anti-HA magnetic beads (88836, Thermo Fisher Scientific, USA). The proteins were fractionated on 10% SDS-PAGE and were transferred to nitrocellulose membranes. Immunoblot analyses were performed using anti-Flag antibody (GTX115043, Genetex, CA, USA) and rabbit polyclonal anti-HA antibody (SG77, Thermo Fisher Scientific, USA).

### 2.8. Determination of Viral Yields through Plaque-Forming Assay

A plaque-forming assay was used to determine the viral titer. The BHK21 clone 15 cells were seeded in 12-well plates (4 × 10^4^ cells per well) containing 1 mL of medium, and were incubated overnight. Then, 0.1 mL of aliquots of a serially diluted virus solution was added to ~70%–80% confluent BHK21 cells. After a 6-h adsorption period, the virus solutions were replaced with 0.75% methylcellulose (Sigma) and 2% FBS. The methylcellulose solution was removed from the wells, and the cells were fixed and stained with a naphthol blue-black solution (0.1% naphthol blue-black, 1.36% sodium acetate, and 6% glacial acetic acid) at six days post-infection. The number of plaque forming units (PFU) per milliliter of DENV-2 was then determined [30].

### 2.9. Establishment of Stable Cells Expressing shSERP1

A lentiviral vector expressing short hairpin RNA, pLKO-1 (National RNAi Core Facility, Academia Sinica, Taipei, Taiwan), was applied to knock down the SERP1 expression. The short hairpin RNA (shRNA) targeting SERP1 (5′-CCCGAAGAGAAGGCGTCTGTA-3′) was used to transfect the packaging HEK 293T cells with the helper vectors pCMV-ΔR 8.91 and pMD.G with Lipofectamine 2000 (Invitrogen). The supernatants containing lentiviral particles were harvested, and the viral titers were assayed with a Lenti-X p24 Rapid Titer Kit (Clontech). The HEK-293 cells were infected with lentiviruses (MOI = 10) in medium containing polybrene (8 µg/mL). Three days after infection, the HEK-293 cells were treated with puromycin to derive a pool of puromycin-resistant clones. Knockdown efficiency was assayed using quantitative reverse transcription-polymerase chain reaction (qRT-PCR).

### 2.10. SERP1 Knockout Cells Generated using the Type II Clustered Regularly Interspaced Short Palindromic Repeats (CRISPR) System

The SERP1^−/−^ Huh7.5 cells were generated by the CRISPR system. This system consists of a bacterial CRISPR-associated endonuclease (Cas9) and small guide RNAs (sgRNAs). Endonuclease Cas9 paired with the sgRNA target DNA causes double-strand breaks (DSBs) [31]. Two target sites (5′-GGCGAGTGAGCGAGTCCGAGGG-3′ and 5′-GCCCCTCTGCGTTCCGCAGCCGG-3′) were chosen from SERP1 exon 1, and were then inserted into the sgRNA expression vectors. The co-expression of the Cas9 and sgRNA plasmids, which were introduced into the Huh7.5 cells by electroporation using microporation (Microporator MP100, Digital Bio, Labtech France), induced site-specific DNA cleavage. Consequently, the SERP1 protein was disrupted at the endogenous level. Individual clones were isolated from the Huh7.5 cell populations. Genomic DNA was extracted using a GeneJET Genomic DNA Purification Kit (Thermo Fisher Scientific). The SERP1 region was amplified at exon 1 by PCR, using the forward primer 5′-GCCAGTTCTCTTCCTCCTGC-3′ and reverse primer 5′-TCACAGGTCTCCCCTTCCGT-3′. The PCR products were cloned into the pGEM-T Easy (Promega) vector and were sequenced. The total RNA was isolated from the Huh7.5 knockout sublines (SERP1^+/−^ and SERP1^−/−^) using an RNeasy Mini Kit (Qiagen, Hilden, Germany), and the RNA was treated with an RNase-Free DNase set (QIAGEN; cat. no. 79254) at 30 °C for 30 min to remove potential DNA contamination (Promega, Madison, WI, USA). Reverse transcription was performed using a RevertAid H minus First Strand cDNA Synthesis Kit (Thermo Scientific). The SERP1 region was amplified from exon 1 to exon 3 by PCR, by using the forward primer 5′-AACGCGCACGCGCA-3′ and reverse primer 5′-ATCCTGATACTTTGAATAATC T-3′. The RT-PCR products were sequenced.

### 2.11. Measurement of SERP1 RNA Levels in the Knockdown Cells and Knockout Sublines by qRT-PCR

The total RNA was isolated from the parental Huh7.5 cells, parental HEK-293 cells, knockdown cells, and knockout sublines using an RNeasy Mini Kit (Qiagen, Hilden, Germany), and the RNA was treated with an RNase-Free DNase set (QIAGEN, cat. no. 79254) at 30 °C for 30 min to remove the contaminating DNA from the RNA preparations. Reverse transcription was performed using a RevertAid H minus First Strand cDNA Synthesis Kit (Thermo Scientific). The specific primers for SERP1 were designed (forward primer: 5′-CGAGCCGAGCCTCGCAGCGGCTC-3′ and reverse primer: 5′-ACCAGGGTCCTACAGACGCCTTC-3′). The expression level of SERP1 mRNA was determined by quantitative real-time PCR (ViiA 7 Real-Time PCR System, Life Technologies) using KAPA™ SYBR^®^ FAST qPCR Kit Master Mix (2×) Universal (Kapa Biosystems). The levels of β-actin mRNA were used as an internal control (forward primer: 5′-TGGATCAGCAAGCAGGAGTATG-3′ and reverse primer: 5′-GCATTTGCGGTGGACGAT-3′) [32]. The amplification program consisted of 40 cycles (each cycle is 95 °C for 3 s, annealing at 60 °C for 30 s) after 95 °C for 3 min. Fluorescent products were detected at the last step of each cycle. The relative quantitative values of SERP1 were normalized to the expression of β-actin. The 2^−△△^*^C^*^T^ method was used to calculate the relative changes in the SERP1 expression.

### 2.12. Transient Replication Activity Assay of the DNA-Launched DENV-2 Replicon

The plasmid pTight-DENV-2 is a DNA-launched infectious clone of DENV2 strain 16681. A DNA launched replicon, pCMV-DV2Rep, derived from the pTight-DENV-2-16681 plasmid, expressed Capsid protein (C), humanized Renilla luciferase (hRluc), the foot-and-mouth disease virus NS2A (FMDV2A) autoprotease, a neomycin resistance gene (Neo), an internal ribosomal entry site (IRES), C-terminal 24 amino acids of Envelope protein (E24), and NS1–NS5. As a negative control, a replicon with an inactivating mutation (GDD to GAA) in the catalytic site of the NS5 RNA-dependent RNA polymerase (RdRp) was used. A transient replication activity assay was performed to monitor the replication efficiencies of the DENV-2 replicons [33,34]. The Huh7.5 cells (parental cells, SERP1-overexpressing cells, and SERP1 knockout cells) were plated in 24-well plates (2 × 104 cells per well) and were co-transfected with 0.1 μg pCMV-DV2Rep and 0.1 μg pTET-OFF using Lipofectamine 2000 (Invitrogen), according to the manufacturer’s protocol. At 24 h post-transfection, doxycycline was added to the culture medium at a final concentration of 2 µg/mL to turn off the DENV-2 gene expression driven by the CMVmin promoter. The luciferase activity was measured using a GloMAX 20/20 luminometer (Promega).

## 3. Results

### 3.1. Identification of Host Factor SERP1 as a DENV-2 NS4B-Interacting Protein

DENV-2 NS4B is one of the membrane-bound viral replication complexes in the ER [14]. The membrane-based split-ubiquitin yeast two-hybrid (MYTH) system was used to characterize the PPIs involving integral membrane and membrane-associated proteins. We decided to utilize NS4B as a bait to screen a liver cDNA library, and identified the ER protein SERP1. Using SERP1 as the prey, we examined the pairwise interactions between SERP1 and NS2B, NS4A, or NS4B. In the MYTH system, the interaction between the bait and prey results in the different degree of activation of the auxotrophic reporter genes, and allows for yeast growth on the selective medium. Ten-fold serial yeast dilutions were spotted onto nonselective plates (lacking tryptophan and leucine (-Trp–Leu)) and selective plates (lacking tryptophan, leucine, histidine, and adenine (-Trp–Leu–His–Ade)) for the detection of protein–protein interactions. As shown in the Figure 1A, the yeast cells co-transformed with SERP1, and either empty vector p-BT3, NS2B, NS4A, or NS4B all grew well on the nonselective plate. However, only the yeast cells transformed with SERP1, and NS4B grew well on different dilutions of the selective plate. It suggested the preferentially binding of NS4B to SERP1 by the MYTH method.

To further validate the interaction, immunofluorescence staining and a NanoBiT complementation assay were applied. The subcellular localization of NS4B and SERP1 was explored using confocal laser scanning microscopy to determine the intracellular distribution of NS4B and SERP1. The Flag-SERP1-overexpressing Huh7.5 cells were infected with DENV-2 (MOI = 5) and fixed at 3 d.p.i. The cells overexpressing Flag-SERP1 were immuno-stained with an anti-Flag antibody localized to a reticular network with an apparent foci throughout the cytoplasm (Figure 1B, upper and middle panels). The Flag-SERP1 protein was distributed around the nucleus. These foci were also immuno-stained with anti-NS4B antibody (Figure 1B, upper and lower panels) and anti-calnexin antibody (Figure 1B, middle and lower panels) in the corresponding cells. The DENV-2 NS4B protein accumulated in the perinuclear region of the cytoplasm, and overlapped with the ER-localized calnexin (Figure 1B, lower panel) in the cells overexpressing SERP1. A quantitative colocalization analysis was performed with the colocalization tool provided with Leica SP5 software. SERP1 was highly colocalized with NS4B and calnexin, with Pearson coefficients of *r* = 0.91 ± 0.02 and *r* = 0.94 ± 0.02, respectively (Figure 1C). Similarly, the colocalization of NS4B was also observed for calnexin (*r* = 0.89 ± 0.03).

To further confirm the intracellular colocalization of SERP1 and DENV-2 NS4B, a NanoBiT complementation assay was performed. For the NanoBiT PPI assay, an optimal combination was selected by testing different combinations of SERP1, NS2B, and NS4B fusion constructs (Figure 1D). Two negative pairs (SmBit-SERP1/NS2B-LgBit and LgBit-PRKAR2A/HaloTag-SmBit) were selected for luminescence detection. In the HEK-293 cells co-transfected with the SmBit-SERP1 and NS4B-LgBit plasmids, the luminescence was up to 30-fold higher than that of the HEK-293 cells co-transfected with the SmBit-SERP1 and NS2B-LgBit plasmids (as a negative control; Figure 1E). The results suggested that SERP1 interacts with DENV-2 NS4B, but not DENV-2 NS2B.

To further confirm the interaction between the SERP1 and the DENV-2 NS4B protein, Flag-tagged SERP1 (pLKO-AS2-Flag-SERP1) was co-expressed in the Huh7.5 cells with HA-tagged DENV-2 NS4B (pLKO-AS2-NS4B-HA; Figure 1F), followed by coimmunoprecipitation and immunoblotting. Equal amounts of protein complexes were precipitated using anti-HA magnetic beads. Co-IP was performed with the anti-Flag antibody and anti-HA antibody. NS4B was shown to be co-immunoprecipitated with SERP1, which indicated that NS4B was associated with SERP1 (Figure 1G, left panel). Equal amounts of protein complexes were analyzed by immunoprecipitation using anti-Flag M2 affinity gel. Co-IP was performed with the anti-Flag antibody and anti-HA antibody. The Western blot analysis revealed that SERP1 was coimmunoprecipitated with DENV-2 NS4B, which indicated that NS4B was associated with SERP1 (Figure 1G, right panel). Thus, the results from the co-IP assays proved that SERP1 interacts with DENV-2 NS4B.

### 3.2. SERP1 Gene Induction by DENV-2 Infection and WT Replicon Transfection

The role of SERP1 in viral infections is poorly characterized. To determine the expression patterns of SERP1 during DENV-2 infection, Huh7.5 cells were infected with DENV-2 (MOI = 1). The SERP1 transcripts were measured by RT-qPCR at various time points. Increasing amounts of SERP1 mRNA were detected in the DENV-2-infected cells during the period of 5 d.p.i., but not in the non-infected cells over time (Figure 2A). A drastic increase (20.7-fold and 34.5-fold at 4 and 5 d.p.i., respectively) in the SERP1 gene expression was observed in the DENV-2-infected cells. The SERP1 gene expressions were not induced in the non-infected cells. To further measure the protein level of SERP1 in the DENV-2 infected cells, we failed to detect the SERP1 protein expression in virus infected cells by either in-house mouse, rabbit, or commercial anti-SERP1 antibodies (Genetex, Abcam, Proteintech Group, Atlas antibodies), possibly because of the poor antigenicity of SERP1 (data not shown). This result indicated that DENV-2 infection induces the expression of SERP1 at the mRNA level at later stages of infection.

To further determine the expression patterns of SERP1 during viral RNA replication, the Huh7.5 cells were transfected with a WT replicon and the mutant replicon (Figure 2B). The SERP1 transcripts were measured by RT-qPCR at various time points. In WT replicon-transfected cells, the expression level of SERP1 increased continuously during the period of 4 d.p.t. (6.1-fold and 19.2-fold at 3 and 4 d.p.t., respectively; Figure 2C). In contrast, SERP1 gene expressions were not induced in mutant replicon-transfected cells. This result indicated that viral RNA replication induces the expression of SERP1 at the replication stage of the life-cycle.

### 3.3. SERP1 Has an Inhibitory Role against DENV-2

The results indicated that the SERP1 expression is upregulated by DENV-2 infection. To further examine the effect of SERP1 on DENV-2 infection, we applied either the overexpression or knockdown of SERP1 in the cells in order to examine the viral yields (titers). To determine the effect of SERP1 overexpression on the viral yields, the Huh7.5 cells stably expressing SERP1 and empty-vector cells were established. Equal amounts of lysates were analyzed by immunoprecipitation using anti-Flag M2 affinity gel. The levels of exogenous recombinant SERP1 proteins were determined by Western blot using anti-Flag and anti-SERP1 antibodies (Figure 3A). The effect of the overexpression of SERP1 on the virus yields was measured by infecting the SERP1-overexpressing Huh7.5 cells and empty-vector cells with DENV-2 at MOI = 1. The viral yields were determined four days after DENV-2 infection. The results indicated that the viral yields in the Huh7.5 cells overexpressing SERP1 were significantly reduced by approximately 37-fold (*p* = 0.02), compared with the empty-vector cells at 4 d.p.i. (Figure 3B).

To investigate the effect of SERP1 silencing on the viral yields, the HEK-293 cells expressing ther shSERP1 and empty-vector cells were established. The RNA interference was used to reduce the cellular levels of SERP1 in the HEK-293 cells. The knockdown of SERP1 resulted in an approximately 51% decrease in the SERP1 mRNA levels in non-infected cells (Figure 3C). The effect of SERP1 silencing on virus yields was measured by infecting the HEK-293 cells expressing shSERP1 and empty-vector cells with DENV-2 at MOI = 1. The results indicated that the knockdown of SERP1 in the HEK-293 cells enhanced DENV-2 yields (3.4-fold; *p* = 0.09), compared with the viral yields in the empty-vector cells at 3 d.p.i (Figure 3D). Taken together, the results suggested that the knockdown of SERP1 in the HEK-293 cells enhanced the DENV-2 yields.

### 3.4. Viral Yields were Significantly Enhanced in SERP1 Knockout Cells

To determine whether the knockout of the SERP1 expression affects the viral yields, the CRISPR/Cas9-mediated deletions of the SERP1 gene were performed in Huh7.5 cells. To verify the engineered deletions in the SERP1 exon 1 in the Huh7.5 cells, RT-PCR using primers flanking the regions of exon 1 and exon 3 were used to amplify (Figure 4A) and sequence (Figure 4B) the two products, 697 bp (wild-type (WT)) and 563 bp (both alleles deleted). The RT-PCR analysis revealed that the truncated RNA in SERP1^−/−^ cell lines resulted in changes in the transcribed RNA. The SERP1 gene encodes a protein with only 66 amino acids. The DNA sequences derived from the reverse transcription of the SERP1 mRNAs transcribed from the mRNAs of SERP1 knock-out cells revealed that there is a 135-nt deletion within SERP1 exon 1, which results in the synthesis of a short peptide (7 amino acids) out-of-frame from SERP1. Thus, the 135-nt deletion completely resulted in knocking out the SERP1 protein, and no synthesis of truncated version of SERP1 protein.

The qRT-PCR analyses revealed that the SERP1 knockout in the Huh7.5 cells resulted in a reduction (>99%) in the mRNA levels of SERP1 (Figure 4C). The WT and SERP1 knockout cells were infected with DENV-2 (MOI = 1). The virus titers were assessed at different time points post-infection. The virus titers from the SERP1 knockout cells were significantly increased (16-fold; *p* = 0.03) at day two post-infection, compared with the titers from the infected WT cells (Figure 4D).

### 3.5. SERP1 Overexpression Significantly Suppresses DENV-2 Viral RNA Replication

The results showed that the viral yields in the Huh7.5 cells overexpressing SERP1 were significantly reduced by approximately 37-fold (*p* = 0.02), compared with the yields in the empty-vector cells (Figure 3B). To further verify that the viral replication step depends on the SERP1 induction to cause virus yield reduction, the DENV-2 replicon containing the luciferase reporter gene was applied to evaluate DENV-2 RNA replication in the Huh7.5 cells (parental cells, SERP1 knockout cells, or SERP1-overexpressing cells). The cells were transfected with WT and mutant replicons (Figure 2B). The luciferase activity of the mutant replicon (Figure 5A) and the WT replicon (Figure 5B) was measured at 1, 2, 3, and 4 d.p.t. The translation efficiency of the DENV-2 viral RNAs was not apparently affected by either the SERP1 knock-out or SERP1 overexpression in the Huh7.5 cells transfected with DENV-2 mutant replicons at 1 d.p.t (Figure 5A). The luciferase activity in the parental cells, SERP1 knockout cells, or SERP1-overexpressing Huh7.5 cells transfected with the replication-defective mutant replicon was almost undetectable at 4 d.p.t (Figure 5A). In contrast, all of the cell lines transfected with the WT replicon displayed an increasing luciferase activity over time. Interestingly, q significantly higher luciferase activity was measured in the SERP1 knockout Huh7.5 cells than in the parental cells at 3 and 4 d.p.t. (*p* < 0.001), whereas the luciferase activity in the Huh7.5 cells overexpressing SERP1 showed a marked decrease compared with that in the parental cells at 4 d.p.t (*p* < 0.001; Figure 5B). Overall, these results supported the findings that SERP1 induction inhibits the viral replication, and causes a reduction in the virus yields. 

### 3.6. Overexpression of NS4B Significantly Improves DENV-2 RNA Replication in the SERP1-Overexpressing Cells

To further determine the importance of the interaction between NS4B and SERP1 during viral replication, the effect of the transient expression of NS4B on the DENV replicon activity within the stable cells expressing SERP1 was evaluated in the Huh7.5 cells. NS2B-HA, △2K-NS4B-HA (lacking signal peptide 2K), and NS4B-HA plasmids were generated, and viral proteins were HA tagged at the carboxyl terminus (Figure 6A). To confirm the expression levels of NS2B or NS4B, the parental cells and SERP1-overexpressing Huh7.5 cells were transfected with a NS2B-HA, △2K-NS4B-HA, or NS4B-HA plasmid, along with the WT DENV-2 replicon plasmid. Equal amounts of cell lysates were analyzed by immunoprecipitation using anti-HA magnetic beads, and the HA-tagged proteins were detected by Western blot using an anti-HA antibody. The results indicated that all of the viral protein expression levels in the parental cells and SERP1-overexpressing Huh7.5 cells were similar (Figure 6B). The luciferase activity derived from the DENV-2 replicon expression in the parental cells either without or with the NS2B, △2K-NS4B, or NS4B expression displayed a similar kinetic pattern from 24 to 96 h.p.t (Figure 6C). Consistent with the results in Figure 5B, the DENV-2 replicon activity within the SERP1-expressing Huh7.5 cells transfected with the DENV-2 replicon was generally lower than that within the parental Huh7.5 cells transfected with the DENV-2 replicon (Figure 6C,D). Interestingly, the overexpression of NS2B-HA or △2K-NS4B-HA in the parental cells transfected with the DENV-2 replicon did not apparently affect the DENV-2 replicon activity pattern, compared with that in the SERP1-overexpressing Huh7.5 cells transfected with the DENV-2 replicon alone (Figure 6D). In contrast, a drastic rescue of the DENV-2 viral replication was found in the SERP1-overexpressing Huh7.5 cells co-transfected with NS4B and the DENV-2 replicon, compared with the SERP1-overexpressing Huh7.5 cells transfected with the DENV-2 replicon alone at 96 d.p.t (*p* < 0.001). These results suggest that the inhibitory effect of SERP1 on DENV-2 replicon activity is alleviated by the overexpression of the NS4B protein, presumably through the interaction between NS4B and SERP1.

## 4. Discussion

In the present study, the host factor SERP1 was identified as a DENV-2 NS4B-interacting protein. The SERP1 expression was induced upon DENV-2 infection and viral RNA replication in the Huh7.5 cells. The knockdown and knockout of SERP1 cells increased the viral yields in the DENV-2-infected cells. The SERP1 overexpression inhibited the viral yields at the viral replication step of the viral life-cycle in the Huh7.5 cells. The overexpression of NS4B in the SERP1-expressing Huh7.5 cells rescued the viral RNA replication inhibited by SERP1 overexpression. The results indicated that the inhibitory effect of SERP1 on the DENV-2 RNA replication is alleviated by the NS4B protein, possibly through the NS4B–SERP1 interaction. We proposed a hypothetical model in which DENV-2 infection induces ER stress and results in the induction of SERP1 expression. The DENV-2 NS4B synthesis occurs during viral RNA replication, and may counteract the inhibitory effect of SERP1 on DENV-2 viral replication via the interaction of NS4B with SERP1 during DENV-2 infection (Figure 7A,B).

The SERP1 expression was greatly induced in the Huh7.5 cells either infected with DENV-2 or transfected with DENV-2 replicon at 4–5 d.p.i, whereas the SERP1 expression was very low in the cells without DENV-2 infection (Figure 2A). The SERP1 induction phenomenon was previously shown to occur in response to various stresses (e.g., hypoxia, ischemic, high glucose, and ER stress response). For example, astrocytes exposed to hypoxia show an increase in SERP1 transcripts in a time-dependent manner, and an approximately 5.7-fold induction of SERP1 antigen in a rat ischemic brain [21]. The SERP1 expression is also induced in cells by treatment with tunicamycin, an endoplasmic reticulum stress inducer [35]. Both the SERP1 transcript and antigen levels are enhanced in the cells under high-glucose conditions [22]. The results suggest that SERP1 may serve as a general stress-induced protein upregulated by various stresses.

The molecular mechanism by which the overexpression of SERP1 inhibits DENV-2 viral RNA replication in Huh7.5 cells is not clear. Little is known regarding the upstream ER stress signaling responsible for SERP1 induction. Our observation supports an important role for SERP1 expression during DENV-2 infection. For example, a drastic reduction in viral yields was found in the DENV-2-infected cells overexpressing SERP1. In contrast, viral yields significantly increased in the SERP1^−/−^ cells infected with DENV-2 (Figure 4D). ER stress induction in the cells infected by DENV has been suggested to modulate virus replication during the virus life-cycle [36]. Previous studies have indicated that the XBP1 knockout in MEFs results in a reduction in several UPR genes, SERP1, IRE1, ERdj4, p58IPK, EDEM, PDI-P5, and HEDJ, confirming their XBP1 dependency [35]. These results suggest that XBP1 is able to regulate the SERP1 expression, implying that SERP1 may belong to the IRE1-XBP1 UPR signaling pathway. Interestingly, the IRE1^−/−^ and congenic MEF cell lines displayed 10-fold lower DENV viral titers, whereas the XBP1^−/−^ congenic MEFs result in no apparent differences in the DENV titers. These results suggest a redundancy in this part of the UPR pathway, and imply that SERP1 is not solely regulated by XBP1. Further studies are needed to illustrate the role and regulation of SERP1 in the UPR signaling pathway.

The DENV-2 replicon activity at 96 h post transfection was severely reduced in the SERP1-overexpressing Huh7.5 cells (Figure 6D). Interestingly, the inhibitory effect of SERP1 on DENV-2 RNA replication in the Huh7.5 cells overexpressing SERP1 was alleviated by the overexpression of the NS4B protein. In contrast, the DENV-2 replicon activity in the Huh7.5 cells gradually increased from 48–96 h post-transfection in the Huh7.5 cells either co-transfected without or with the NS2B or NS4B protein (Figure 6C). Specifically, overexpressing the NS4B protein in the parental Huh7.5 cells moderately increased the RNA replication compared with the Huh7.5 cells transfected only with DENV-2 replicon. We envisioned that it is likely that the amounts of endogenous SERP1 derived from the induction by the DENV-2 replicon replication are less than those of the SERP1 from stable cells overexpressing SERP1. The expression level of SERP1 determines the degree of inhibitory effect on the DENV-2 RNA replication. This phenomenon implied that NS4B may have different roles, in addition to being a component of the DENV-2 replication complex at the ER during DENV-2 replication. 

What is the molecular mechanism by which NS4B counteracts the inhibitory effect of SERP1 on DENV-2 RNA replication? NS4B overexpression may induce cellular responses to prevent the antiviral activity derived from SERP1. A previous study indicated that NS4B is involved in interferon inhibition in DENV-infected cells, in order to prevent the establishment of a cellular antiviral state to enhance virus yields and replication [13]. Huh7.5 cells exhibit a defect in IFN induction by dsRNA, because of a single point mutation in the dsRNA sensor retinoic acid-inducible gene-I (RIG-I) [37]. Therefore, it is unlikely that IFN is responsible for the inhibitory effect of the SERP1 expression on DENV replication. DENV NS4B has also been identified as an important organizer of the membranous DENV replication complex, and as an inducer of intracellular membrane rearrangements [14], which may result in cellular signaling to counteract the inhibitory effect of SERP1 expression on DENV replication. Alternatively, it is anticipated that the role of NS4B in facilitating replication may be as a SERP1 binder to counteract the antiviral role of SERP1. This scenario is supported by the observation that the DENV-2 NS4B–SERP1 interaction was suggested by the results of the MYTH system analysis, subcellular colocalization experiments, NanoBiT complementation assay, and co-immunoprecipitation assay (Figure 1). However, the other possibility cannot be excluded that SERP1 impairs DENV-2 RNA replication via interrupting the interaction between NS4B and NS2B-NS3, as DENV-2 NS4B was reported to interact with NS2B-NS3, and the interaction is crucial for viral RNA replication [38]. The overexpression of NS4B alleviates the inhibitory effect of SERP1 on DENV-2 RNA replication, possibly by restoring the binding of NS4B to NS2B-NS3 within the replication complex. Our previous attempt to show the interaction between SERP1 and NS2B-NS3 failed, because of the toxicity of the NS2B-NS3 overexpression in yeast, Huh7.5, and HEK-293 cells. Thus, whether SERP1 overexpression affects the interaction between DENV-2 NS4B and NS2B-NS3 cannot be evaluated in the cells overexpressing NS2B-NS3. Further studies are needed in order to evaluate the possibility. Given the fact that NS4B is a key component of the DENV-2 replication complex located in the ER, NS4B may play multiple roles in counteracting antiviral activity during DENV-2 replication, by interacting with various viral proteins and host factors.

## 5. Conclusions

In summary, human SERP1 was identified as a potential partner of DENV-2 NS4B. SERP1 expression was greatly induced in DENV-2-infected cells. The knockdown and knockout of SERP1 markedly enhanced viral yields and replication. In contrast, the overexpression of SERP1 resulted in the inhibition of viral yields and RNA replication, whereas the inhibitory effect of SERP1 on the DENV-2 RNA replication was relieved by the NS4B overexpression. It is anticipated that there are complicated interactions between DENV-2 NS4B and SERP1. The NS4B–SERP1 interaction may delicately modulate DENV-2 replication during the DENV life-cycle. It would be interesting to know whether SERP1 interacts with NS4B from other DENV serotypes, although the amino acid sequences NS4B among four DENV serotypes display a moderate degree of conservation. Further studies are needed in order to determine the broad spectrum effect of SERP1 on DENV or other flavivirus replications. These results may support the beneficial effect of SERP1 on DENV or flavivirus, which may provide clues for the development of effective SERP1 inducers for therapeutic purposes.

## Figures and Tables

**Figure 1 viruses-11-00787-f001:**
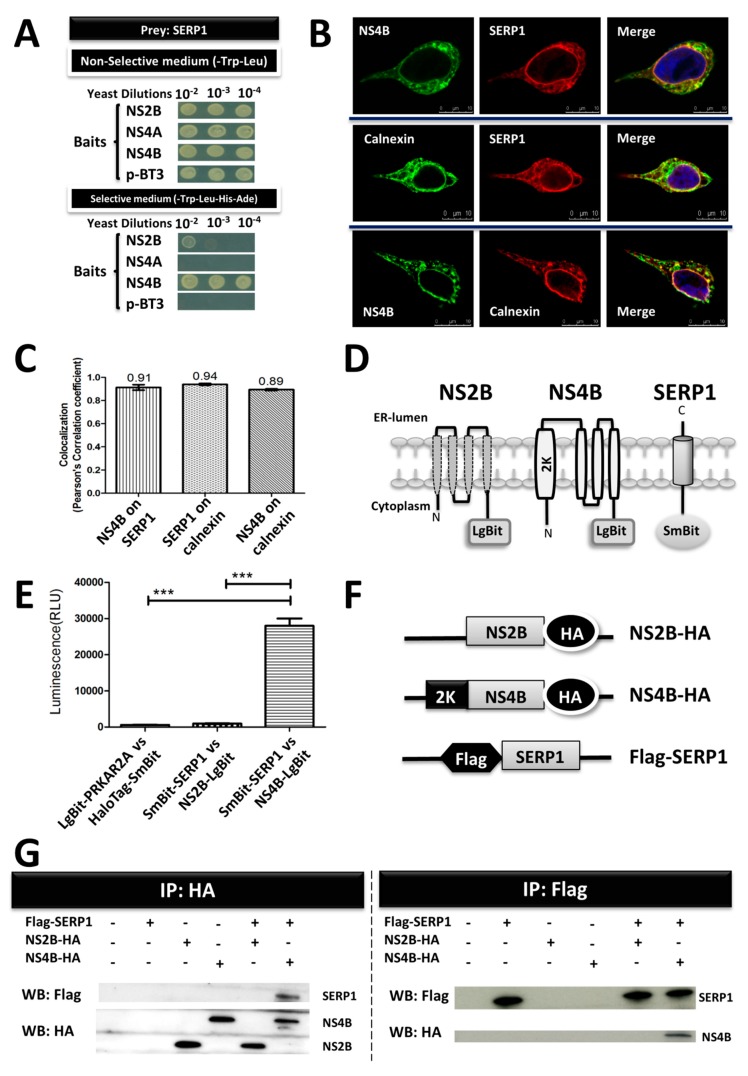
Protein–protein interactions between stress-associated endoplasmic reticulum protein 1 (SERP1) and nonstructural protein (NS)4B. (**A**) NS4B interacts with SERP1, as shown in a membrane-base split ubiquitin yeast-two-hybrid assay. Yeast was co-transformed with the baits p-BT3 NS2B, p-BT3 NS4A, p-BT3 NS4B, and p-BT3 (vector only), and the prey pPR3-SERP1. Ten-fold serial yeast dilutions were spotted onto nonselective plates (-Trp–Leu; lacking tryptophan and leucine) and selective plates (-Trp–Leu–His–Ade; lacking tryptophan, leucine, histidine, and adenine) for the detection of protein–protein interactions. (**B**) Dengue virus type 2 (DENV-2) NS4B colocalized with SERP1 and the endoplasmic reticulum (ER) marker calnexin. The Flag-SERP1-overexpressing Huh7.5 cells were infected with DENV-2 (multiplicity of infection (MOI) = 5) and subjected to co-stain with antibodies raised against Flag, NS4B, or ER-located calnexin. At 72 h post-infection, the subcellular distributions of SERP1, NS4B, or calnexin were examined by indirect immunofluorescence assay using the corresponding antibodies. NS4B and SERP1 showed a strong colocalization (upper panel). ER-located calnexin and SERP1 (middle panel) and NS4B (lower panel) also showed colocalization. The nuclei were stained with DAPI (blue). Scale bar, 25 μm. (**C**) The colocalization analysis between SERP1, NS4B, and calnexin was quantified using Pearson’s correlation coefficients. Each counterstain was determined for 10 Huh7.5 cells in three independent experiments using the colocalization tool provided, with Leica-SP5 software. The values are shown as the average of the Pearson’s correlation coefficients in 10 cells. Error bars indicate the means ± standard errors of the mean (SEMs). (**D**) The topological scheme of the SERP1, NS4B, and NS2B complexes with selected interaction pairs in a Nano-Luc Binary Technology (NanoBiT) protein–protein interactions (PPIs) assay. (**E**) The interactions of SEPR1-NS4B and SEPR1-NS2B were determined in the HEK-293 cells using a NanoBiT complementation assay. Luminescence is expressed as the means ± SEMs from three independent experiments (*n* = 3). ***, *p* < 0.001 (Student’s *t*-test). (**F**) Schematic diagram of the NS2B-HA (pLKO-AS2-NS2B-HA), NS4B-HA (pLKO-AS2-NS4B-HA), and Flag-SERP1 (pLKO-AS2-Flag-SERP1) fusion constructs. (**G**) The interaction of SEPR1-NS4B was determined in the Huh7.5 cells by co-immunoprecipitation (co-IP). The Huh7.5 cells were transfected or co-transfected with pLKO-AS2-Flag-SERP1, pLKO-AS2-NS2B-HA, and pLKO-AS2-NS4B-HA. Equal amounts of protein complexes were analyzed by immunoprecipitation using anti-Flag M2 affinity gel or anti-HA magnetic beads. The co-IP was performed with the anti-Flag antibody and anti-HA antibody.

**Figure 2 viruses-11-00787-f002:**
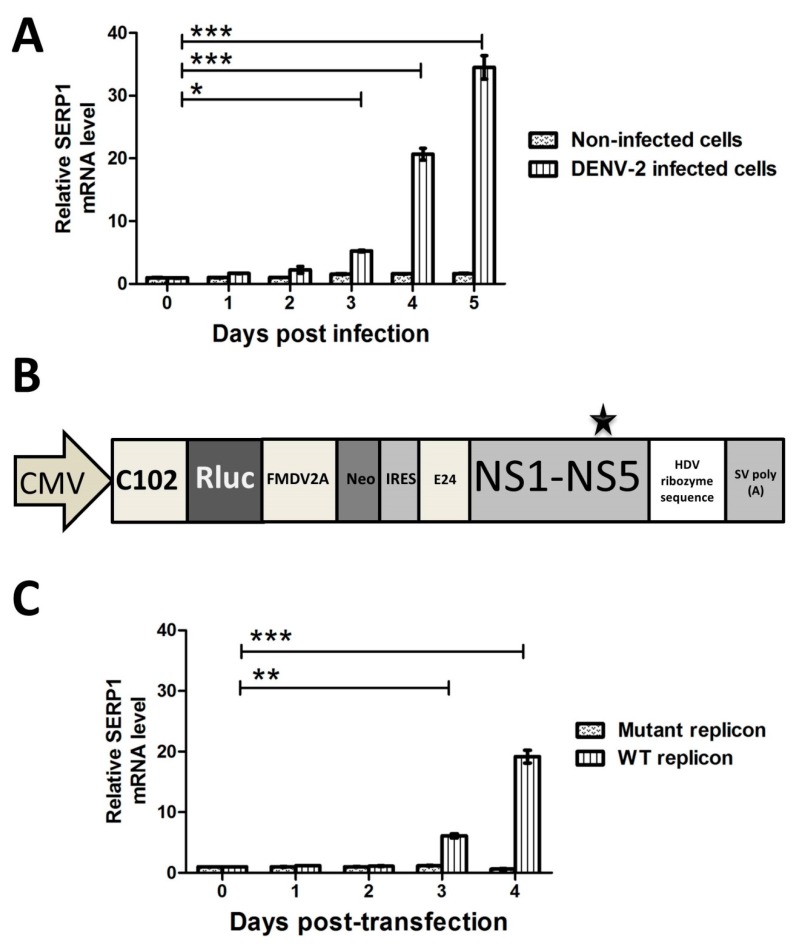
SERP1 expressions were induced in Huh7.5 cells upon DENV-2 infection and wild-type (WT) replicon transfection. (**A**) The Huh7.5 cells were uninfected or infected with DENV-2 at an MOI = 1. The quantification of SERP1 transcripts was performed by RT-qPCR at 0, 1, 2, 3, 4, and 5 d.p.i. The relative quantitative values of the SERP1 gene were normalized to the level of β-actin. *, *p* < 0.05; ***, *p* < 0.001 (Student’s *t*-test). (**B**) Schematic representation of the DNA-launched DENV-2 reporter replicon pCMV-DV2Rep, which was used in a transient replicon assay. The transcriptional expression of the replicon RNA is under the control of the CMVmin promoter, and the 3′ terminus of the transcript is processed by hepatitis delta virus (HDV) ribozyme sequences. The N-terminal 102 amino acids of the C protein (C102), the Renilla luciferase gene (Rluc), the FMDV2A cleavage site, a neomycin resistance gene (Neo), an internal ribosome entry site (IRES) element, the C-terminal 24 amino acids of E (E24), the entire NS protein region (NS1 to NS5), the HDV ribozyme sequence, and the SV40 poly(A) signal sequence are indicated. The star represents the replication-defective mutant—a replicon with an inactivating mutation (Gly–Asp–Asp (GDD) to Gly–Ala–Ala (GAA) at the amino acids 662–664 of NS5) in the catalytic site of the NS5 RNA-dependent RNA polymerase (RdRp) was used as a negative control. (**C**) The Huh7.5 cells were transfected with the WT replicon or mutant replicon. The quantification of the SERP1 transcripts was performed by RT-qPCR at 0, 1, 2, 3, and 4 d.p.t. The relative quantitative values of the SERP1 gene were normalized to the level of β-actin. **, *p* < 0.01; ***, *p* < 0.001 (Student’s *t*-test).

**Figure 3 viruses-11-00787-f003:**
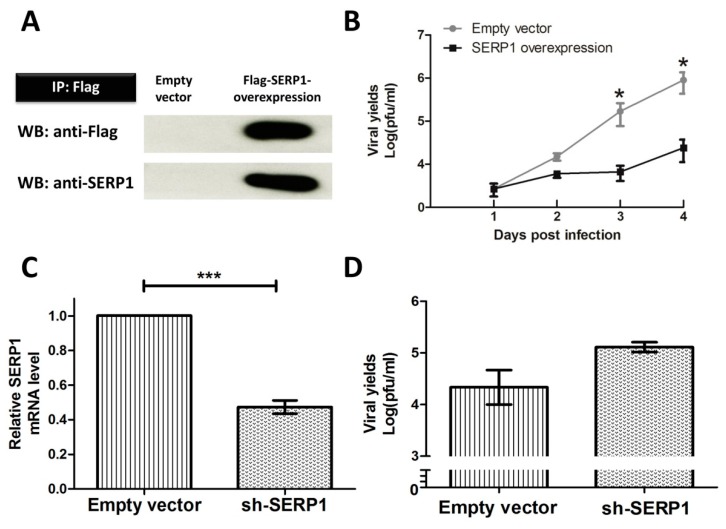
The SERP1 overexpression inhibited DENV-2 infection, and SERP1 knockdown increased DENV-2 infection. (**A**) Western blot analysis of Flag-tagged SERP1 proteins in Huh7.5 cells expressing exogenous SERP1 and empty-vector cells. Equal amounts of lysates were incubated with anti-Flag M2 affinity gel, and the precipitates were analyzed by Western blot using the anti-Flag antibody and anti-SERP1 antibody. (**B**) The reduction in DENV-2 yields in SERP1-overexpressing cells infected with DENV-2. We established Huh7.5 cells with a stable expression of SERP1 and empty-vector cells. The cells were infected with DENV-2 at MOI = 1. The virus yields were determined at the indicated times. Infectious virus yields in the BHK21 clone 15 cells were quantified by plaque assay. Tde differences in the virus yields between the Huh7.5 cells stably expressing SERP1 and empty-vector cells at 3 or 4 d.p.i. were analyzed using Student’s *t*-test. *, *p* < 0.05. (**C**) A RT-qPCR analysis of the SERP1 mRNA expression in the empty vector HEK-293 cells and knockdown cells. shRNA-mediated knockdown of SERP1 reduces the mRNA expression in the HEK-293 uninfected cells. The expression values were normalized to the β-actin expression. The values are the means ± standard errors of the means (SEMs). ***, *p* < 0.001 (Student’s *t*-test). (**D**) The knockdown of the SERP1 cells increased the viral yields of DENV-2. We established HEK-293 cells with a SERP1 knockdown by a specific shRNA. The HEK-293 cells stably expressing shSEPR1 were infected with DENV-2 at MOI = 1. The virus yields were determined at 3 d.p.i.

**Figure 4 viruses-11-00787-f004:**
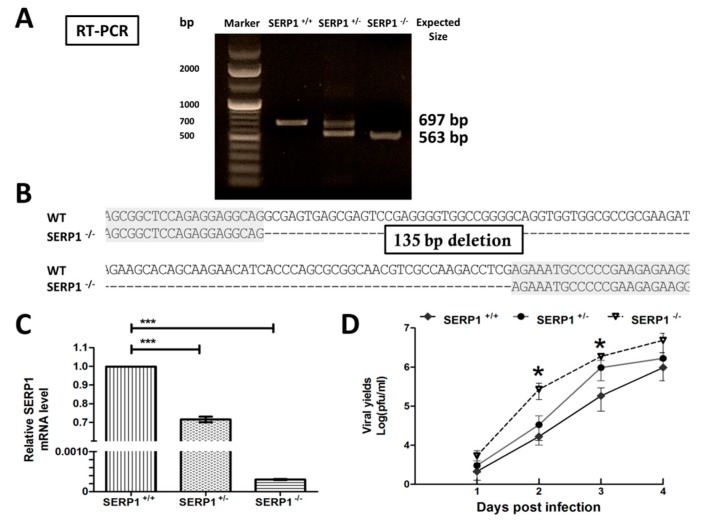
Knockout of SERP1 by the II clustered regularly interspaced short palindromic repeats (CRISPR)/Cas9 system in Huh7.5 cells decreased SERP1 mRNA levels, and viral yields were significantly enhanced in the SERP1 knockout cells. (**A**) SERP1 RNA expression patterns in the SERP1 knockout sublines. The products of RT-PCR performed on RNA isolated from the parental Huh7.5 cells (SERP1^+/+^) and SERP1 knockout sublines (SERP1^+/−^ and SERP1^−/−^) using the primers in SERP1 exon 1 and exon 3, which generate a 697 bp product. The SERP1 mutant allele was amplified as a 563 bp product, where the SERP1 exon 1 region was deleted. (**B**) Sequencing analysis of the parental Huh7.5 cells and SERP1 knockout cell lines from the RT-PCR product. (**C**) Detection of SERP1 mRNA levels in the parental Huh7.5 cells and SERP1 knockout sublines by qRT-PCR. The expression values are normalized to β-actin expression. ***, *p* < 0.001 (Student’s *t*-test). (**D**) Kinetics of DENV-2 replication in the parental cells and SERP1 knockout sublines. The parental Huh7.5 cells and SERP1 knockout sublines were infected with DENV-2 (MOI = 1) at the indicated times. Infectious virus yield in the BHK21 clone 15 cells was quantified by plaque assay. The differences in virus yields between the parental cells (SERP1^+/+^) and SERP1 knockout cells (SERP1^−/−^) at 2 or 3 d.p.i. were analyzed using Student’s *t*-test. *, *p* < 0.05.

**Figure 5 viruses-11-00787-f005:**
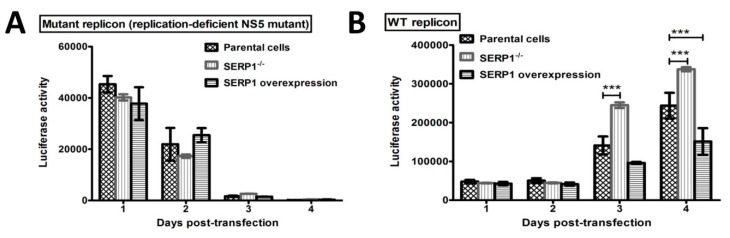
The effect of SERP1 on the DENV-2 replicon capacity in the Huh7.5 cells. The replication activities of the transient expression of DNA-launched mutant replicon (**A**) or WT replicon (**B**) plasmids were detected in the parental cells, stable cells overexpressing SERP1, and SERP1 knockout cells. The luciferase activity of the transfected cells was measured at 1, 2, 3, and 4 d.p.t. The error bars represent the SEMs from three independent experiments. The differences in the luciferase activity between the transfected cells at 3 or 4 d.p.t. were analyzed using Student’s t-test. ***, *p* < 0.001 (relative to the parental cells).

**Figure 6 viruses-11-00787-f006:**
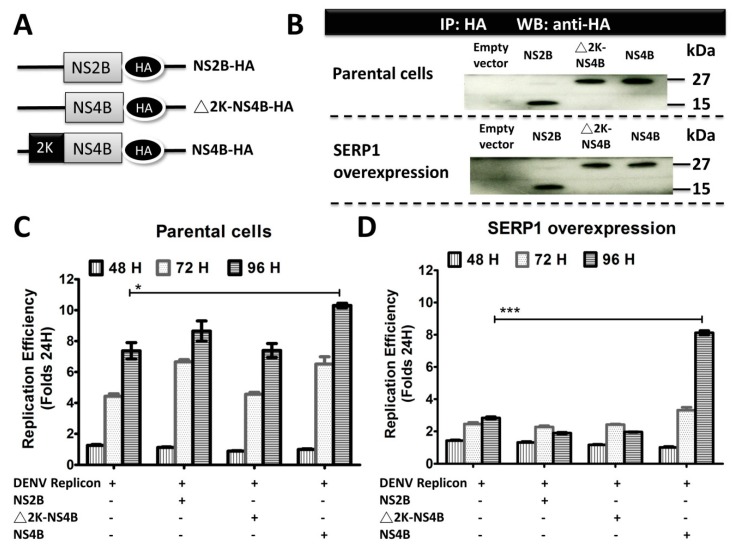
Overexpression of NS4B improves the virus replication in the Huh7.5 cells overexpressing SERP1. (**A**) Schematic diagram of HA C-terminal fusion constructs of NS2B and NS4B. The N-terminal 2K signal peptide was deleted (△2K-NS4B-HA). (**B**) Immunoblot analysis of HA-tagged NS2B and NS4B proteins in the parental cells and SERP1-overexpressing Huh7.5 cells. All of the fragments represented in panel (**A**) were cloned in pLKO-AS2 to tag the C-terminal end of each protein. Equal amounts of lysates were incubated with anti-HA magnetic beads, and the precipitates were analyzed by Western blot using an anti-HA antibody. The parental cells (**C**) and SERP1-overexpressing Huh7.5 cells (**D**) were co-transfected with WT replicon and plasmids encoding individual viral proteins, as indicated. The cell lysates were harvested 24, 48, 72, and 96 h after transfection, and the luciferase activity of the transfected cells was measured. The replication efficiency was calculated by determining the ratio of luciferase activity obtained at 48, 72, and 96 h, to the average value obtained from all of the replicon constructs at 24 h post-transfection. The error bars represent the standard errors of the means (SEMs) from three independent experiments. *, *p* < 0.05; ***, *p* < 0.001 (Student’s *t*-test).

**Figure 7 viruses-11-00787-f007:**
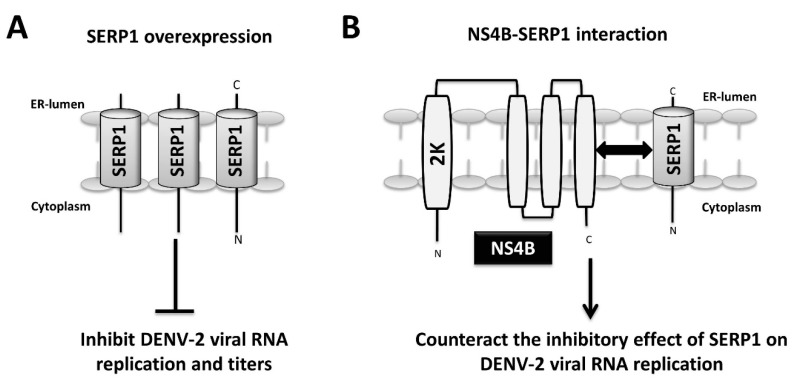
Hypothetical model of SERP1-mediated DENV-2 infection. (**A**) SERP1 overexpression inhibits DENV-2 viral RNA replication and titers. (**B**) DENV-2 NS4B interacts with SERP1. DENV-2 NS4B may alleviate the inhibitory effect of SERP1 on DENV-2 viral replication via the interaction of NS4B with SERP1.

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
