# Peer review of "A Dengue Virus Type 2 (DENV-2) NS4B-Interacting Host Factor, SERP1, Reduces DENV-2 Production by Suppressing Viral RNA Replication"

_viruses, 2019, doi:10.3390/v11090787_

Round 1

Reviewer 1 Report

Reviewer:  In the resubmitted manuscript of Tian et al described experiments where DENV-2 strain 16681 was used to identify SERP1 from Huh7.5 cells as a binding factor for the viral NS4B protein.

The new version is significantly improved.

Minor points:

Reviewer:   Lane 2-4: The title of the manuscript still implies that SERP1 is suppressing viral RNA replication for Dengue viruses in general. Please change to :

                   ….., Reduces Dengue Virus 2 Production by Suppressing ……

Lane 60:     NS4B, from which DENV subtype the NS4B was used regarding to the (17) reference?

Lane 306:   Fig. 1A. To make the figure more understandable. Are the dilutions “Yeast dilutions”. If yes, please change it to “Yeast Dilutions”.

Fig. 1 E:     Relative Light Units: RUL should therefore be RLU.

                   From the Material & Method section it is not clear if it is RLU or RLU/sec. Please specify.

Lane 333    ENV-2 should be DENV-2

Lane 365    (GDD to GAA), please specify if it is aa or na. Is GAA Gly-Ala-Ala?

Lane 426    Fig. 4A, please change Ladder to for example to Marker

Lane 493    Text: (A) Schematic diagram of NS2B and NS4B-HA fusion constructs.

                   Both proteins are fused to HA Is it: Schematic diagram of HA C-terminal fusion constructs of NS2B and NS4B.

                   The following sentence can therefore be deleted.

Author Response

Reviewer 1

Minor points:

Q1: Lane 2-4: The title of the manuscript still implies that SERP1 is suppressing viral RNA replication for Dengue viruses in general. Please change to :

                   ….., Reduces Dengue Virus 2 Production by Suppressing ……

Response: As suggested, we revised the title on Ln3-4.

A Dengue Virus (Type 2) NS4B-interacting Host Factor, SERP1, Reduces Dengue Virus 2 Production by Suppressing Viral RNA Replication

Q2: Lane 60: NS4B, from which DENV subtype the NS4B was used regarding to the (17) reference?

Response: DENV-2/16681 NS4B protein

As suggested, we revised the description on Ln61.

Q3: Lane 306:   Fig. 1A. To make the figure more understandable. Are the dilutions “Yeast dilutions”. If yes, please change it to “Yeast Dilutions”.

Response: As suggested, we revised the Fig. 1A.

Q4: Fig. 1 E:   Relative Light Units: RUL should therefore be RLU.

From the Material & Method section it is not clear if it is RLU or RLU/sec. Please specify.

Response:

Relative Luminescence Unit (RLU)

As suggested, we revised the Fig. 1E and Ln175.

Q5: Lane 333  ENV-2 should be DENV-2

Response: As suggested, we revised the description on Ln333.

Q6: Lane 365 (GDD to GAA), please specify if it is aa or na. Is GAA Gly-Ala-Ala?

Response: As suggested, we revised the description on Ln365.

Q7: Lane 426  Fig. 4A, please change Ladder to for example to Marker

Response: As suggested, we revised the Fig. 4A.

Q8: Lane 493 Text: (A) Schematic diagram of NS2B and NS4B-HA fusion constructs.

Both proteins are fused to HA Is it: Schematic diagram of HA C-terminal fusion constructs of NS2B and NS4B.

The following sentence can therefore be deleted.

Response: As suggested, we revised the description on Ln494-496.

Schematic diagram of HA C-terminal fusion constructs of NS2B and NS4B. The N-terminal 2K signal peptide was deleted (△2K-NS4B-HA)

Reviewer 2 Report

Authors essentially revised the manuscript well to address all comments. However, regarding to response to major comment #4, revision did not reflect the comment.

Lines 571-580; author added the discussion on the inhibition mechanism of SERP1. However, author misunderstood the comment. NS4B was reported to interact with NS3 and this interaction is essential for viral replication. In this manuscript, author observed inhibitory effects of SERP1 on DENV2 replication by interacting with NS4B, and overexpression of NS4B could improve DENV2 replication. The comment meant that SERP1 might interrupt NS4B-NS3 interaction by interaction with NS4B. Interaction of SERP1 with NS4B should result in reduced NS4B-NS3 interaction, resulting in inhibition of DENV2 replication. Under the NS4B overexpression condition, the abundant NS4B could interact with NS3, resulting in improved replication, even if the some portion of NS4B was used to interact with SERP1. Thus, if author can demonstrate that NS4B-NS3 interaction was reduced under SERP1 overexpression condition, it should be very interesting. Please re-consider the discussion, although technical difficulty of this experiment is understandable.

Author Response

Reviewer 2:

Authors essentially revised the manuscript well to address all comments. However, regarding to response to major comment #4, revision did not reflect the comment.

Lines 571-580; author added the discussion on the inhibition mechanism of SERP1. However, author misunderstood the comment. NS4B was reported to interact with NS3 and this interaction is essential for viral replication. In this manuscript, author observed inhibitory effects of SERP1 on DENV2 replication by interacting with NS4B, and overexpression of NS4B could improve DENV2 replication. The comment meant that SERP1 might interrupt NS4B-NS3 interaction by interaction with NS4B. Interaction of SERP1 with NS4B should result in reduced NS4B-NS3 interaction, resulting in inhibition of DENV2 replication. Under the NS4B overexpression condition, the abundant NS4B could interact with NS3, resulting in improved replication, even if the some portion of NS4B was used to interact with SERP1. Thus, if author can demonstrate that NS4B-NS3 interaction was reduced under SERP1 overexpression condition, it should be very interesting. Please re-consider the discussion, although technical difficulty of this experiment is understandable.

Response: We apologize to fail to make it clear in the discussion. We revised and showed on Ln574-583.

This manuscript is a resubmission of an earlier submission. The following is a list of the peer review reports and author responses from that submission.

Round 1

Reviewer 1 Report

In this study, a host protein, SERP1, was identified to interact with DENV NS4B, and an inhibitory role of SERP1 on DENV replication was found. Although these findings are interesting and important to the field, the current study has left a lot of gaps that limits the strength of the conclusion.

1. The key questions of this study are: (1) Whether SERP1 inhibits DENV replication through interaction with NS4B? (2) The nonstructural proteins of flavivirus always form a replication complex, so whether SERP1 interacts with other proteins in replication complex? Additional experiments are needed to address these questions. 

2. The authors did the screen for host proteins interacting with DENV NS2B and NS4B by using yeast two-hybrid system. Given that NS2B functions as a co-factor with NS3 protease, it’s unclear why the authors did the screen for NS2B but not NS2B-NS3.

3. To further confirm the interaction of NS4B and SERP1, a co-immunoprecipitation assay should be performed.  

4. Figure 2, the mRNA level of SERP1 should be measured in both DENV-infected and uninfected cells and a comparison between two groups is needed. And the protein levels of SERP1 should also be examined.

5. Figure 3, it is better to determine the overexpression and knock-down of SERP1 by western-blot using anti-SERP1 antibody.

6. It is hard to understand why NS4B can improve DENV RNA replication only in SERP1-overexpressing cells but not in parental cells, because the overexpressed NS4B would also inhibit the function of endogenous SERP1. This should at least be discussed.

Reviewer 2 Report

Reviewer:  In the manuscript of Tian et al described experiments where DENV-2 strain 16681 was used to identify SERP1 from Huh7.5 cells as a binding factor for the viral NS4B protein in contrast to NS2B. They used direct virus-cell infection assays, protein-protein binding assays or studied genome replication using a 16681 based replicon.

Overall the manuscript is very well written and most of the experiments are explained in great detail.             

Unfortunately, the study concept is limited on the DENV-2--Huh7.5 interaction, lacking any control experiment with other Flavi- or Dengue viruses or cell lines.

The authors should consider the following points to improve their study:

Major points:

Reviewer:   Lane 2-4: The title of the manuscript implies that SERP1 is suppressing viral RNA replication for Dengue viruses in general. This is clearly not shown since only DENV-2 strain 16681 or a 16681 based replicon was used. The authors can therefore:

 (i)  support this statement by adding experiments for other Dengue viruses or NS4B variants.

                   or

 (ii) should change the title and the main message of the manuscript for the observation that SERP1 is binding to the DENV-2/16681 NS4B in Huh7.5 cells. The term DENV must therefore generally replaced by DENV-2 and the discussion should also mention these experimental restrictions.

Reviewer:   Lane 268, Fig. 1A: From the figure legend and the method described in section 2.2 it is difficult for the reader to interpret the dot results since it is not clear what the dots represent. The authors cited a reference (#29) that is not open access. Therefore, it would be appreciated when the method would be described in more detail and also more understandable to enable the reader to evaluate the experimental read out.

Minor points:

Reviewer:   Lane 88: is SERP1 a novel protein?                  

Reviewer:   Lane 90: should be ……Cells stably overexpressing SERP1………..

Reviewer:   Lane 90: have you shown inhibition of viral RNA replication or was at the replicon?

Reviewer:   Lane 112-113: The amino acid shortcuts and explanations for -Leu etcetera should be also given in the figure legend of Fig. 1A.

Reviewer:   Lane 214: 1668 should be 16681

Reviewer:   Lane 297: was the assay done always at the same time on each of the 5 days? If not it would be days after post infection

Reviewer:   Lane 320: should be: …. that the knockdown of SERP1 in Huh7.5 cells enhanced DENV-2 yields.

Reviewer:   Lane 322, Fig. 3A: please add the explanations for WB and especially IP in the figure legend. Same for Fig. 6B.

Reviewer:   Lane 335: …. Stable cells were infected…… Do the authors mean:  Cells stably expressing shSERP1 were infected….

Reviewer:   Lane 498-499: The data do not support such a general statement.

In the literature, inhibitors for NS4B are described and NS4B can escape from these inhibitors by antigenic variation. How can the authors be sure that this is not possible for the SERP1 binding activity of NS4B? The authors should mention these points in the discussion section. Especially genetic differences of NS4B in the DENV group,

Reviewer 3 Report

Dengue virus (DENV) has been spread throughout tropical and subtropical regions, and causes public health problems. It is estimated that 390 million DENV infections occur annually worldwide. Although one chimeric tetravalent vaccine was licensed, no specific therapeutic anti-viral drug is available yet. This manuscript revealed that the ER stress-induced SERP1 interacts with DENV NS4B protein and inhibits DENV replication. This finding is important to understand a novel mechanism in the cells to combat with DENV infection and may provide a novel insight for anti-DENV drug development. However, the manuscript is required minor modification.

Major comments

1.       Have you checked if SERP1 is induced when DENV NS5 mutant replicon is introduced into cells? If RNA replication is required for SERP1 induction, SERP1 level is not increased by the DENV NS5 mutant replicon. Thus, author can verify the trigger for the SERP1 induction (viral protein or RNA replication).

2.       Relating above, Fig 5B revealed that translation of viral protein is not affected by SERP1. Please mention this fact in the Result and/or Discussion.

3.       Fig4; Do the SERP1 knock-out cells still produce truncated version of SERP1? If yes, have you confirm that the truncated SERP1 does NOT interact with NS4B?

4.       Regarding to the mechanism of the inhibition of DENV replication by overexpression of SERP1, one possible explanation is that the SERP1 compete NS3-NS4B interaction. Since NS4B was reported to interact with NS3 and NS4B-NS3 interaction was required for RNA replication (Chatel-Chaix L et al.A Combined Genetic-Proteomic Approach Identifies Residues within Dengue Virus NS4B Critical for Interaction with NS3 and Viral Replication.J Virol. 2015 Jul;89(14):7170-86.), Interaction of SERP1-NS4B may reduce NS4B-NS3 interaction. Please discuss this possibility. If author can confirm that overexpression of SERP1 reduced NS4B-NS3 interaction, resulting in inhibited RNA replication, the impact of this manuscript is elevated.

Minor comments

1.       Fig 3C; Is this data obtained from infected cells or uninfected cells?

2.       Fig 3C; Have you checked SERP1 protein level?

3.       Have you tested other DENV serotypes?

4.       Line116; DENV should read DENV2.

5.       Line214; strain 1668 should read strain 16681